# It was tough, but necessary. Organizational changes in a community based maternity care system during the first wave of the COVID-19 pandemic: A qualitative analysis in the Netherlands

**Iris F. Appelman**[1]*, **Suzanne M. Thompson**[2], **Lauri M. M. van den Berg**[2], **Janneke T. Gitsels van der Wal**[2], **Ank de Jonge**[2], **Martine H. Hollander**[1]*

1 Department of Obstetrics and Gynaecology, Radboud University Medical Center, Nijmegen, The Netherlands, 2 Department of Midwifery Science, AVAG/Amsterdam Public Health, Amsterdam University Medical Centre, Vrije Universiteit Amsterdam, Amsterdam, The Netherlands

* iris.appelman96@gmail.com (IFA); martine.hollander@radboudumc.nl (MHH)

**Data Availability Statement:** Due to confidentiality restrictions by the ethical committee of Amsterdam

## Abstract

### Introduction

The Coronavirus SARS-CoV-2 pandemic necessitated several changes in maternity care. We investigated maternity care providers' opinions on the positive and negative effects of these changes and on potential areas of improvement for future maternity care both in times of crisis and in regular maternity care.

### Methods

We conducted nineteen semi-structured in-depth interviews with obstetricians, obstetric residents, community-based and hospital-based midwives and obstetric nurses. The interviews were thematically analysed using inductive Thematic analysis.

### Results

Five themes were generated: '(Dis)proportionate measures', 'A significant impact of COVID-19', 'Differing views on inter-provider cooperation', 'Reluctance to seek help' and 'Lessons learnt'. The Central Organizing Concept was: 'It was tough but necessary'. The majority of participants were positive about most of the measures that were taken and about their proportionality. These measures had a significant impact on maternity care providers, both mentally and on an organizational level. Most hospital-based care providers were positive about professional cooperation and communication, but some community-based midwives indicated that the cooperation between different midwifery care practices was suboptimal. Negative effects mentioned were a higher threshold for women to seek care, less partner involvement and perceived more fear among women and their partners, especially around birth. The most significant positive effect mentioned was increased use of

UMC, Vrije Universiteit, the datasets used and/or analysed during the current study are available upon reasonable request to the data manager of the department of Midwifery Science, Lauren Ancion, l.f.ancion@amsterdamumc.nl.

**Funding:** The study has been funded by the Royal Dutch Organization of Midwives (KNOV), and AdJ received this funding. The KNOV had no involvement of the funders in the study design, data analyses, manuscript preparation, and publication decisions. Website: https://www.knov.nl/.

**Competing interests:** The authors have declared that no competing interests exist.

eHealth tools. Recommendations for future care were to consider the necessity of prenatal and postnatal care more critically, to replace some face-to-face visits with eHealth and to provide more individualised care.

## Conclusion

Maternity care providers experienced measures and organizational changes during the first wave of the COVID19-pandemic as tough, but necessary. They believed that a more critical consideration of medically necessary care, increased use of e-health and more individualised care might contribute to making maternity care more sustainable during and after the pandemic.

## Introduction

The Coronavirus SARS-CoV-2 pandemic has had a major effect on the organization of maternity care worldwide. When the number of COVID-19 cases in the Netherlands increased excessively, several measures were implemented by the Dutch government to protect the inhabitants against the virus [1, 2]. Examples of these measures are social isolation, such as a lockdown and a curfew, to prevent people from spreading the virus [2].

In maternity care as well, measures were taken to protect women, neonates, care providers, and the maternity care system [3]. There have been international signals that these changes had a significant impact on both women and maternity care providers [4]. To understand the impact of these changes on maternity care, it is important to know how maternity care was organized before the COVID-19 pandemic. In the Netherlands, low-risk pregnant women, which is the majority of pregnant women in the Netherlands, receive care from a community midwife. Hospital based midwives are part of the medical team working under the supervision of obstetricians, together with obstetric residents. They see women who develop risk factors or complications after they have been referred by community midwives, in case the pregnancy of a woman becomes high risk. In 2008, the Royal Dutch Organization of Midwives (KNOV) outlined a blueprint for prenatal care in the community care setting. This blueprint stated that women see a midwife on average thirteen times during a full-term pregnancy. Most of these visits, on average nine, are in the third trimester [5, 6]. The blueprint also recommends two ultrasounds: one to determine gestational age and one mid-pregnancy ultrasound to check for congenital anomalies [5].

During the first two months of 2020, the Dutch government did not communicate any concerns about the Coronavirus 'SARS-CoV-2' reaching the Netherlands. In addition, Dutch virologists and epidemiologists reassured the population through social media that this was not likely to become a serious problem in this country. However, in the second half of February, concerning news reports from Italy reached Dutch (social) media channels, and suddenly the threat of an outbreak in the Netherlands seemed much more realistic. On February 27th 2020, the first Dutch patient tested positive for the virus [1]. After this, events followed quickly one after another. The first confirmed death occurred on March 6th [7]. Initially, testing was restricted to seriously ill patients, which contributed to an underestimation of the speed with which the virus was spreading. The advent of carnival in the south of the country, combined with Dutch people returning from skiing holidays in the heavily affected parts of Italy, caused a spiralling number of infections, with an epicentre in the southern province of Brabant.

On March 9th, the first measures were installed by the national government: no more hand-shakes, washing your hands as many times as possible and sneezing in the crook of your elbow [2]. In the weeks that followed, many more measures were decreed, both nationwide and through hastily formed taskforces in hospitals. News bulletins from hospital crisis teams followed quickly and frequently, with sometimes several new directives per day. The first measures instated by national and regional maternity care crisis teams were aimed at keeping the number of contacts between maternity care providers and patients, as low as possible. One of these changes was reducing the number of prenatal visits in primary care from an average of thirteen to seven for a full-term pregnancy while keeping the two ultrasounds [3]. These seven prenatal visits were deemed to be essential. Some other prenatal visits were replaced by telephone or online consultations. In addition, women were contacted by telephone before visiting the midwifery practice to check if they had COVID-19 related symptoms. They received information about hygiene guidelines and other measures, and their social and medical situation was discussed over the telephone [3]. One other notable change was that women were obliged to attend visits and ultrasounds alone, without a partner or other family member [3].

In 2018, the Dutch Organization for Midwives outlined basic postnatal care in the community, including a minimum of four home visits by the midwife during the first ten days postpartum [8]. The COVID-19 guideline drawn up by the Dutch Organization for Midwives advised that postpartum home visits were recommended only for strict medical or social reasons and all other forms of contact should be through phone calls or 'window visits' [3]. In the Netherlands, women are entitled to postpartum maternity care in their own home during the first eight days following childbirth, a maternity care assistant monitors the health of both mother and new-born and supports them in tasks such as feeding and bathing [8].

The aforementioned changes in maternity care were not only visible in the community midwifery setting but also in hospitals. Hospitals, likewise, instituted a reduction in the number of prenatal visits and an increase in the use of e-health, such as consultations using technology like Zoom and Skype [9]. Additionally, women were requested to attend consultations and ultrasounds unaccompanied, and during labour there was only one other person, often the partner, allowed in the room [9]. Finally, some hospitals instituted a policy of banning outpatient births under supervision of community midwives, leaving low risk women no other options than to have a homebirth or be referred to secondary care, which is visible in the amount of homebirths [10].

To date, several studies have investigated experiences of maternity care providers' experiences with specific changes in maternity during the COVID-19 pandemic, such as the increase of e-Health [11, 12]. Furthermore, several studies show that care providers' emotional health was under pressure [13, 14]. However, as far as we know, there have not been any studies that investigate the overall experiences of maternity care providers with the changes in maternity care during the first wave of the COVID-19 pandemic. To understand the experiences of maternity care providers during a situation as unique as the first wave of the COVID-19 pandemic in depth, both quantitative and qualitative data are needed. The quantitative data provided insight in the scale of the problem, and may suggest causes and effects, as can be seen in this recently published study [15]. However, qualitative analysis is needed to answer questions as to why certain things occurred/were done. In this current study, we aimed to investigate which policy changes in maternity care during the first COVID-19 wave were perceived as positive by care providers and could offer future improvements. In addition, we also wanted to know which changes were perceived as unfavourable by care providers and therefore needed re-evaluation in case of another crisis. Finally, we were interested in evaluating cooperation between different maternity care providers, practices and hospitals.

## Methods

### Research team and reflexivity

The interview team consisted of a medical student (IA), an obstetrician (MH) and a midwife (ST). The interviews with the obstetricians and obstetric residents were conducted by MH, and the interviews with the midwives by ST. During the interviews, IA was present to record the conversations. All of the authors have an affinity with maternity care, and are working in maternity care in different functions, for example as a registered midwife or gynaecologist, or are in another way involved in maternity care in the Netherlands. Both MH and ST have a background in maternity care and are familiar with the Dutch system. MH works in maternity care during COVID-19, which could have influenced the questions that she asked. ST was not practicing midwifery during the COVID pandemic but was active in education and research. The interviewers are both experienced in qualitative research. Because both interviewers are maternity care professionals, they also had personal opinions about the measures taken in maternity care during the first wave of the pandemic, which may have influenced the initial topic list. Findings were discussed regularly within the research team to reflect on the initial themes and on the role of the interviewers.

### Study design and participants

This study is part of a larger multidisciplinary study, the WAAG -study. The WAAG-study is a mixed-methods study in which the experiences of both maternity care providers and clients with the changes in maternity care during the first wave of the COVID-19 pandemic have been investigated.

For another part of the WAAG-study, a digital questionnaire was developed in which maternity care providers' experiences with the changes in maternity care were examined through multiple-choice and open questions [16]. At the end of the questionnaire, participants were asked if they consented to being contacted for an in-depth interview. If they answered 'yes', their name, postal code of their primary workplace and email address were added to a list.

We aimed to include a sample of all relevant categories of respondents, to cover different professional experiences during the first wave of the pandemic. We therefore included five obstetricians, two obstetric residents, seven community-based midwives, two hospital-based midwives and two obstetric nurses (Table 1). The researchers selected the potential partici- pants based on residence, profession, and echelon (community-based or University/non-uni- versity hospital-based). We purposefully included more community based midwives and

**Table 1. Sampling plan.**

| N | Profession | Region | Further characteristics of interest |
|---|---|---|---|
| 7 | Community- based midwife | South (epicentre of the coronavirus in the Netherlands | |
| | | West/ Randstad (including Utrecht) | |
| | | North | |
| | | East | |
| 2 | Hospital- based midwife | South (epicentre of the coronavirus in the Netherlands) | Working in a University Hospital |
| | | North or West or East | Working in a non-University hospital |
| 7 | Obstetrician/ obstetric resident (distribution 5/2) | South (epicentre of the coronavirus in the Netherlands | Working in a University Hospital |
| | | West/Randstad (including Utrecht | Working in a non-university hospital |
| | | North | |
| | | East | |

obstetricians because they carry final responsibility for the care of women in their practice or hospital. Some of the participants were known by name to the interviewers through professional and social networks, but none had a professional relation with the interviewers, which could have influenced the results.

## Data collection

The interviews were conducted during June and July 2020 through online video meetings with Skype for Business or Microsoft Teams by MH or ST. IA was present to record the interviews. The interviews were semi-structured by use of a topic list (S1 Appendix). These topics were based on the aforementioned questionnaire and its answers. This list was adjusted throughout the study based on new topics that were identified during the interviews. Interviews were recorded through an encrypted audio device and transcribed verbatim. Audio files and transcripts were stored in a password protected university database.

## Data-analysis

In our study, a Thematic Analysis was used to analyse the interviews [17]. Qualitative data analysis software program Atlas.ti was used for coding [18]. The coding process was started with reading through all the transcripts carefully to gain a deep understanding of the content. Next, inductive coding was performed. New codes were added as new interviews were analysed. The codes were grouped into subthemes and merged into themes by IA and MH (S2 Appendix). MH coded one interview at the beginning of the analysis and ST coded two interviews halfway through the process to check for any missing codes. Finally, the themes were tested on all the interviews by LB. Differences of opinion regarding the themes were resolved through discussion between the authors.

## Ethical considerations

This study was submitted to the Medical Ethics Review Committee of the VU University Medical Centre (reference number 2020.255) which confirmed that the Medical Research Involving Human Subjects Act (WMO) did not apply and therefore an official approval of the study by the committee was not required. All participants gave verbal and written informed consent and had the opportunity to withdraw at any point during the study without consequences.

## Results

A total of forty-seven community-based midwives, four hospital-based midwives, two obstetric nurses, sixteen obstetricians and three obstetric residents were contacted by email. Not all were available at short notice due to the summer period. In total, we interviewed seven community-based midwives, two hospital-based midwives, six obstetricians, two obstetric residents and two obstetric nurses. The participants' characteristics are shown below (Table 2).

Data saturation was reached after fifteen interviews. We confirmed this with four additional interviews during which no new codes were identified. Following data analysis, five major themes were generated: '(Dis)proportionate measures', 'A significant impact of COVID-19', 'Differing views on inter-provider cooperation', 'Reluctance to seek help' and 'Lessons learnt'. After careful consideration of all data, one Central Organizing Concept was identified: 'It was tough but necessary'. It is important to note that none of the respondents worked on COVID wards.

**Table 2. Characteristics.**

|  | Characteristics | N |
|---|---|---|
| Gender | Male | 3 |
|  | Female | 16 |
| Profession | Obstetrician | 6 |
|  | Gynaecology resident | 2 |
|  | Hospital-based midwife | 2 |
|  | Community-based midwife | 7 |
|  | Obstetric nurse | 2 |
| Work region | North | 5 |
|  | East | 4 |
|  | South | 5 |
|  | West | 5 |

## (Dis)proportionate measures

The subjects that participants name in the interviews can roughly be divided into two domains that reflect childbirth: changes in antenatal and postnatal care, and changes in intrapartum care. The most significant change in antenatal and postnatal care in hospitals and midwifery practices was the downscaling of regular maternity care. Many participants believed that women did not like these measures and these measures increased fear and concern among pregnant women and, therefore, negatively impacted their wellbeing. since they missed some elements of care they would have liked to have had.

> 'The restrictive measures during the first part of pregnancy care—I think I can say this for sure—was something that our clients experienced as annoying. Not so annoying that they moan about it, more that they find it a pity. And many [clients] found it concerning to miss that part of care.'

(Community-based midwife 5, region South)

Several participants indicated that the impact of the COVID-19 pandemic on hospitals was not particularly noticeable for them because the labour ward was set apart from the rest of the hospital, and there were not many infected patients on the maternity wards. All participants thought it was essential for routine intrapartum care to continue as usual. To achieve this, most obstetricians insisted early on in the pandemic that maternity care providers should remain employed in their department rather than be seconded to a COVID-department.

An important change in intrapartum care in some hospitals was related to the possibility of outpatient birth with a community-based midwife. In some regions, women were refused the option of birth as an outpatient. In other hospitals, however, outpatient birth was possible, but only without the presence of the woman's community-based midwife, who would, under normal circumstances, be responsible for outpatient birth. According to maternity care providers, both of these measures caused uncertainty and displeasure among women and midwives.

> 'One thing that had a major impact was when the hospital (COVID) taskforce decided to suspend out-patient births in the hospital. These are births where the woman chooses for hospital, without a medical indication. This caused a bit of a commotion. . .also women

under the care of the obstetricians asked whether they would still be able to give birth in hospital [. . .]'

(Obstetrician 2, region East, non-University hospital)

Overall, most participants were content with the proportionality and timeliness of the measures.

*'I am satisfied with it. I think they [the measures] were introduced on time, but not too early. [. . .] And when things escalated, everything was turned into a crisis structure very fast. And that worked very well at that moment. [. . .] So I think that everyone accepted the measures because they noticed that it (the situation) was dealt with adequately and quickly. And I also thought that it was proportional.'*

(Obstetrician 2, region East, non-University hospital)

There were, however, regional differences visible. Some professionals in the North of the Netherlands, where the pandemic arrived later and was less severe, experienced the measures as too strict. At the time of the interviews, the official position of the Dutch government was still that there was no asymptomatic spread of the virus.

*'Constantly disinfecting your hands, even though you've given good instructions to people with symptoms, well, that was disproportional. Here in the north, it wasn't as severe, it wasn't spreading around as much. But to not go on home visits when they [the woman and her family] have no symptoms, that isn't OK. And no visitors to the maternity ward, also not OK, that was disproportional'*

(Community-based midwife 2, region North)

Others thought that the measures' timeliness was inadequate and that the measures should have been introduced earlier. Most participants also spoke about maternity care's relaxing of restrictions when the first peak of the COVID-19 pandemic was over. Several participants thought that this should have happened faster, although many participants indicated that it was understandable that it took some time because there were still concerns about problems that hospitals lacked the staff to manage the delivery rooms at full capacity and time was needed for care providers to recover. An important recommendation made by the participants was that, in the case of a second wave of COVID-19 (or another pandemic), maternity care should not be downscaled quite so quickly and severely. On the one hand, some participants believed that downscaling was the right thing to do. On the other hand, they felt that it should be done more gradually in a subsequent wave, should this occur.

'We won't be scaling down (care) like idiots again. We panicked in the down scaling of care. . .all training, conferences on hold, all elective care on hold. Now we are upscaling carefully, downscaling should also be done carefully. That was an important lesson.' (Obstetrician 1, region South, non-University hospital)

## A significant impact of COVID-19

The COVID-19 pandemic had a significant impact on maternity care providers. Most providers, in particular community-based midwives, indicated that it was an overwhelming and uncertain period in which they experienced challenging events in both their professional and personal lives. Some participants stated that there was such a high proportionality routine on

them, that they did not have time to recharge their energy. The work burden was mainly increased because of sickness or mandatory self-isolation of maternity care workers, which created a higher work burden for the others. Also, some maternity care workers indicate that the higher burden came from making protocols for COVID-19 care. Several participants indicated that they experienced the most uncertainty during the beginning of the pandemic, when much was still unknown about COVID-19 as a disease. Most care providers expressed little concern about their own well-being but were mainly concerned about others' health in their immediate surroundings.

*'Well, fear for the condition itself, not so much because, maybe a bit naïve but I thought; I am young, I am healthy and it will not spread that fast, it will be alright. Personally, I wasn't that afraid of it.'*

(Obstetrician 2, region East, non-University hospital)

However, a few participants indicated that colleagues had serious concerns about becoming infected.

*'There were a lot of concerns about infections, a lot of concerns.[. . .] Nurses especially, and some doctors, a few people older than 60 as well, they were a bit anxious but they continued to work. But the nurses especially, they were very anxious.'*

(Obstetrician 3, region West, non-University hospital)

The participants also experienced a number of organizational effects of the pandemic on their work. Obstetricians and obstetric residents indicated an increase in working from home and changes made to duty schedules. One such change was that obstetric residents, in particular, were seconded to work in other departments, such as COVID-19 departments, while the obstetricians had to do more hands-on work themselves. Most hospital-based participants reported a great deal of flexibility in their department when it came to the duty roster, which created a feeling of enhanced team spirit. Overall, maternity care providers that worked in a hospital-based setting were positive about the organizational effects of the pandemic on their work.

*'[. . .]There was an enormous feeling of unity in the hospital and a huge willingness among all people (staff) to do what was necessary. [. . .] everyone dropped everything and when there was a shift open, within ten minutes I had ten people who were willing to cover it. So you immediately had a feeling that we were all in this together.'*

(Hospital-based midwife 1, region South, non-University hospital)

Some community-based midwives, however, were less positive. They indicated an increase in organizational burden and that their work activities changed from primarily patient/client-oriented to making rules and protocols, which caused a decrease in job satisfaction among some.

*'[. . .] We greatly reduced client contact as soon as the advice was given to do so. But that means that behind the scenes, there is a whole lot more organizational work to do to organize the practice. Consultations with other practices in the neighbourhood about how to set up care. [. . .] The organizational burden was very much increased in the past couple of months while the amount of client contact was drastically reduced.'*

(Community-based midwife 5, region South)

Besides the effects on maternity care providers, some participants also perceived an effect on the relationship between the parents -to-be and themselves. They stated that it was more of a challenge to establish a trusting relationship with the women and their partners, even though that is of great importance according to the midwives. The establishment of a trustful relationship and the impact that COVID-19 had on this was not mentioned by the obstetricians and obstetric residents in our study. Most participants also indicated that they perceived negative effects of the measures taken on (the expectant) mother and her partner. The most frequently mentioned effect was an increase in fear and uncertainty, especially about their unborn child. All maternity care providers that worked in a hospital-based setting stated that only the partner was allowed to accompany the woman in the hospital, excluding other family members or childbirth support such as doulas. The maternity care providers that worked in a hospital-based setting in our study felt this led to uncertainty for the women, as these changes impacted women's expressed wishes for their childbirth experience. Another issue was that new mothers could not visit their child on the neonatology ward if they were suspected of having of COVID-19, which the participants felt was upsetting for all concerned.

Some community-based midwives stated that they felt that there was less partner involvement because partners were excluded from antenatal visits and ultrasounds, which caused the partner to be less involved in the pregnancy than usual.

*'[. . .] for example, this week I was present at a birth, it was the couple's third child. The husband. . . said; 'hey, this is actually the first time that I hear the baby's heartbeat.' [. . .] he hadn't heard the heartbeat since the twentieth week of pregnancy and now she [the woman] is in labour. Yes, that was kind of tough for me, that kind of hit me.'*

(Community-based midwife 6, region East)

## Differing views on inter-provider cooperation

The changes in maternity care organisation had an effect on both cooperation and communication on two levels: within each (care) echelon and across echelons (transmural). Overall, most of the participants indicated that they experienced a strong feeling of unity and an enhanced team spirit during the first wave of the COVID19-pandemic, both within their echelon as well as transmural. They stated that maternity care providers experienced a feeling of getting through this tough period together, even though some maternity care providers were not on the same page when it came to specific policies.

There are, however, notable differences between the experiences of obstetricians, obstetric residents and hospital-based midwives, and the experiences of community-based midwives. The hospital-based clinicians described acceptable cooperation and communication within their echelon, with a good hospital atmosphere and a high level of commitment from colleagues. The community-based midwives in this study were content with the cooperation amongst colleagues in their practice, but were less positive about the cooperation and communication with other midwifery practices. This could be due to the fact that hospitals did not compete with each other for patients, whereas several midwifery practices may operate in the same area and are therefore in competition with each other for clients. Almost all community-based midwives indicated differences in policy between different community-based midwifery practices in their region. For example, in some practices, pregnant women were allowed to bring their partner to a clinic visit or ultrasound appointment, contrary to the national guidelines. According to the participants, this led to women 'shopping' for midwifery care between midwifery practices, increasing feelings of competition between community midwifery practices as a result. The experienced competition was a cause of friction between community-

based midwives. Some of the community-based midwives mentioned that competition between their own and other midwifery practices, existent before the pandemic, increased due to COVID-19.

*'[. . .] but also, the regions are very small and there is obviously a lot of competition so every time you hear; 'oh, but my friend who lives 10 kilometres away could do this and that, and I cannot do that here with you.'. [. . .] In other regions, there is already quite some downscaling, whereas here that is very limited.'*

(Community-based midwife 5, region South)

Opinions about transmural (between hospital-based providers and community-based midwives) cooperation and communication were overall positive, especially among obstetricians and obstetric residents. Most maternity care providers who worked in a hospital-based setting indicated that communication lines between community care, hospital care and the overarching regional and national organizations were effective. Almost all participants indicated that some regional task force was established to communicate both national and regional measures to all maternity care providers and be a point of contact for the hospital departments if there were difficulties. Almost all participants were positive about these taskforces and stated that measures were communicated well, which created a feeling of clarity. In most regional maternity care organizations, a COVID-WhatsApp group was established, to keep everyone updated. The participants who had experience with such a WhatsApp group were very positive.

*'[. . .] I believe in the first week of (the) COVID (pandemic), we had a department meeting and we instantly created a COVID-WhatsApp group. And that worked really well! [. . .] We had very short lines (of communication). And if there was a question, organizational or from the community-based midwives or the paediatricians, we could resolve these very quickly in that COVID-WhatsApp group and the solution would be sent to everyone by their own representative. So that was really handy [. . .].'*

(Obstetrician 4, region South, non-University hospital)

Some obstetricians indicated that there were regional recommendations about early intrapartum transfers from community to hospital. Some hospital-based midwives were negative about these recommendations. The hospital-based maternity care providers experienced that the community-based midwives sometimes referred clients to the hospital too early. These earlier referrals led to a shift in risk selection from community-based providers to hospital-based maternity care providers, resulting in friction between them.

*[. . .] I did more emergency consultations because the community-based midwives saw very few patients themselves. A lot of those consultations were, well, if a woman called the midwife, she'd be sent right away to the hospital. With a lot of these (consultations) I did feel that if the community (based) midwife had seen the woman, she'd have been better able to ascertain if this was a reason for an emergency consultation or a matter of reassurance or treatment in the community. With these [referrals] we noticed an increase in referrals from the community. . .women who would be sent in without being seen by the community midwife first.'*

(Obstetric resident 1, region South, non-University hospital)

### Reluctance to seek help

Most participants indicated that they noticed increased anxiety among pregnant women during the initial stages of the COVID-pandemic. The participants in both community and hospital settings noticed that many women were sometimes reluctant to seek care. All participants described that they noticed that women seemed to have a higher threshold for approaching maternity care providers, perhaps because women feared to visit a health care institution based on the presumption of a higher chance of getting infected. Furthermore, the participants indicated that women did not want to burden health care capacity.

*'[. . .] the number of births that I assisted and the number of home births increased. There were a lot of people who chose, I have to say, against the hospital. They chose a home birth because they did not want to go to the hospital. So that was a negatively motivated choice.'*

(Community-based midwife 7, region West)

*'And that they (the women) did experience a bit of a threshold to call us if there was something wrong, because then we had to come over and they didn't know if we would mind that, while we tried to tell them that we really wanted to come. We said: it really depends on you (the women), if you want a visit, just tell us and we will come. However, people still experienced a threshold and that's bad.'*

(Community-based midwife 3, region West)

Maternity care providers also noticed that this avoidance caused some women to revaluate their choice for place of birth. This re-evaluation may have increased the number of home births, although this has yet to be verified by national data. Another reason that participants mentioned that could have contributed to women choosing a different place of birth is that only the partner was allowed to accompany the woman during labour in the hospital. Others, such as family members or professional childbirth support, e.g. doulas, were excluded. The participants in our study perceived that this measure resulted in uncertainty for the women, because some of them could not have the birth that they had planned beforehand.

### Lessons learnt

Almost all participants in our study indicated that the most positive effect of the COVID-pandemic was the increased use of innovations around e-Health. To keep the number of contacts as low as possible, many face-to-face consultations were replaced by digital consultations. One of the benefits frequently described by all participants is that eHealth is patient-friendly, saving women (travel)time and money, and that it could be more accessible for the pregnant women than a face-to-face visit with the obstetrician.

*'But I do think that it benefits the women, in terms of logistics but also in terms of a feeling of control. You're in your own living room and on the agreed time your tablet goes off and then you have a conversation with the doctor. I can imagine that people feel freer or maybe more equal than when you have to go to the hospital to see THE obstetrician [. . .]'*

(Obstetrician 2, region East, non-University hospital)

Although almost all of the participants in our study were positive about increasing the number of contacts by phone and video consultations, some of them also saw disadvantages to using these forms of communication. One such disadvantage was that non-verbal

communication is less easy to interpret during telephone and video consultations. This disadvantage was mainly mentioned by community-based midwives, who noted the importance of non-verbal communication and placed great value on ascertaining non-verbal communication as part of their care. Most community-based midwives had a conservative attitude to eHealth, tending to maintain contact by phone rather than using other eHealth forms, such as video consultations.

Almost all participants in our study concluded that the increase of contacts by phone and other forms of eHealth was overall positive, but that is important to preselect which antenatal or postnatal visits could be replaced by phone or other forms of eHealth as video consultations, and which ones should remain face-to-face. The participants indicated that the preselection depends on the type of visit but also on the preferences of the woman, and her characteristics, such as women with a language barrier or women with low levels of health literacy.

Another positive effect that some hospital-based clinicians, in particular, indicated was a decrease in the number of visitors in the maternity ward. The decrease of visitors contributed to a more restful environment for women and their babies. Additionally, some participants indicated that, overall, the COVID-19 pandemic might have created conditions that facilitated more rest during pregnancy as women did not have to attend clinic visits as often as usual. Two participants even indicated that they saw a decrease in the number of premature births during the pandemic, which they linked to the possibly more restful environment for the women.

*'[. . .] I am under the impression that, and I also hear this from our neonatologist, that there were less premature births, a lot less premature births. [. . .] But maybe it is true, because on one hand, the pregnant women are restless, but on the other hand, life got very clear and orderly.'*

(Obstetrician 5, region West, non-university hospital)

Almost all participants indicated that the pandemic's initial phase highlighted that standard maternity care consists of many routinized prenatal and postpartum visits and that it might be appropriate to consider tailoring care to the needs of the individual woman. Participants also stated that all maternity professionals should critically examine which aspects of care are essential face-to-face—provided either in the hospital or community midwifery practices—and which aspects of care could be replaced with eHealth.

*'[. . .] in general, in obstetric care, we sometimes do too many consultations. For example, from week 36, we see a pregnant woman every week, which is actually not strictly necessary if she is healthy, and in the past period of time [due to COVID-19], there were less of these consultations. So I think that it would be good to take a critical look at that. How did we experience it, did we notice any problems, how can we replace the consultations, by contacts by phone or other forms of eHealth for example, and how can we bring the amount of clinic visits down.'*

(Obstetrician 1, region South, non-University hospital)

One participant recommended that in a future pandemic or similar crisis, it would be favourable to create designated COVID-hospitals and non-COVID-hospitals, to lower the threshold for women to seek care.

*'I think it is better (to separate hospitals in COVID- and non-COVID care). Because then you will not get the fear. But is it quit difficult, and we will probably not be able to organise the*

*separation of hospitals. Which is unfortunately because I think that the regular healthcare would have felt more safe for people and could have continued better.'*

(Obstetrician 6, region North, non-University hospital)

## Central organizing concept: It was tough, but necessary

The Central Organizing Concept that we identified was evident throughout the majority of the interviews. The participants in our study indicated that the measures were tough, but necessary. In hindsight, some of the measures may have been too strict or unnecessary. However, at the beginning of the pandemic, there was little choice other than to implement them, given the degree of uncertainty about COVID-19 as a disease at the time, particularly for pregnant women.

*'I think that we had no choice in our region, considering that our whole OR (operating room) and recovery and everything was full with patients. [. . .] If you saw how many of our patients were transported to Germany and other parts of the Netherlands, then I understand very well that there was no choice. And afterwards it is always easy of course, but it went so fast, so many patients came in every day. And so many died in the Emergency Department [. . .]. With today's knowledge, everything is easy to reason but it went so exponential and so fast, you had to do something.'*

(Hospital-based midwife 1, region South, non-University hospital)

## Discussion

In this qualitative study, we aimed to explore and gain insight into the experiences of maternity care providers with the organisational changes in maternity care during the COVID-19 pandemic. Five themes were generated; '(Dis)proportionate measures', 'A significant impact of COVID-19', 'Differing views on inter-provider cooperation', 'Reluctance to seek help' and 'Lessons learnt'. These could be synthesized into one Central Organizing Concept; 'it was tough, but necessary'. This Concept captures the essence of the experiences of maternity care providers during the COVID-19 pandemic.

### Proportionality of the measures

The participants in our study were generally content with the proportionality of the measures that were taken. This was found to be in line with the experiences of maternity care providers in other countries [19]. However, the proportionality of the measures appeared to be perceived differently, in different regions of the Netherlands. Participants from the northern part of the Netherlands, for example, perceived the measures to be too strict. This finding was in line with the overall lower incidence of COVID-19 in that part of the Netherlands [20]. It is probable that a 'one size fits all' approach for containing the spread of COVID-19 or future pandemic disease outbreaks is less desirable and that a tailored approach is necessary, such as the 'tier' approach used in the UK [21]. However, this approach could lead to women 'shopping' for care in areas where COVID-19 measures are more flexible than in their own region. This could increase the competition between care providers, as was the case among primary midwifery practices in our study. Many participants highlighted the disadvantages of competition, both for women and care providers during a pandemic crisis. This dilemma should be discussed within and between the professional organisations of both midwives and obstetricians. A future consideration, particularly for professional organisations, should be in calling for

solidarity and unity among midwifery practices and hospitals to abide by national policies in order to reduce friction and competition.

Another aspect of proportionality was reflected in the scaling down of services in all hospitals in order to prepare for COVID-19 patients. One of the participants indicated that setting up separate COVID hospitals like there have been for instance in China and India at the start of the pandemic [22] might be a solution and that, by concentrating COVID care in certain hospitals, other hospitals might be able to continue to offer their usual maternity care. This may lower the threshold for women to contact maternity care providers. However, choosing the hospitals where COVID care is going to be centred might be challenging, because it is difficult to predict which regions of the country will be affected the most. Besides, it might be challenging to find (non-maternity care) staff for the COVID centred care settings, because it is to be expected that most care providers would rather work in their own department than in a COVID hospital.

## Necessity of making changes in the number of visits

One of the measures implemented during this period was a reduction in antenatal clinic visits and ultrasounds for pregnant women. Most participants stated that this measure was one of the reasons that women experienced uncertainty and that there were many concerns, both from a health-related and social perspective. However, several of the participants indicated that this pandemic illustrated that perhaps there are unnecessary visits in the regular maternity care schedule in the Netherlands.

The KNOV developed a guideline for standard prenatal care, which details an average of thirteen prenatal visits for a full-term pregnancy. In the literature, some studies suggest that what women are most concerned with is that their own and their baby's health being checked regularly [23–25]. In the first trimester, women indicate that they are most in need of confirmation of their pregnancy, and information about their pregnancy, childbirth and the postnatal period [26, 27]. These studies appear to illustrate that women desire at least the amount of care that they usually receive, and that some women may require more care than is usually scheduled.

Some studies showed the effects of a reduction in the number of antenatal clinic visits. Dowswell et al. analysed the effect of a reduction in prenatal visits (from twelve to eight visits) in high-income countries [26]. The study showed no significant differences in perinatal mortality between the reduced visits group and the standard visits group [26]. However, there was a slightly increased risk of premature births among the group who received a reduced number of prenatal visits although these results are marginally statistically significant (risk ratio 1.24, 95% CI 1.01–1.51) [26]. Other studies suggest that a lower number of visits may lead to decreased satisfaction with the childbirth experience and poorer socio-emotional outcomes among women [27, 28]. Considering all of the above, it does not seem desirable to reduce the number of antenatal and postnatal clinic visits in regular maternity care. Most of the participants in this study, however, do see benefits in using eHealth for some consultations while conducting others face-to-face and feel that care could be better tailored to meet the needs of women. Future research is needed to explore which of the twelve to fourteen standard antenatal clinic visits are necessary to meet women's needs for care and which others, while being 'nice to have' are not essential. Both medical outcomes and psychosocial aspects of the visits should be explored and, based on these findings, protocols could be made to ensure that every woman gets prenatal care that is acceptable, safe and efficient.

## Possible reduction of premature births

In contrast to the study of Downswell et al. which suggests that a reduction of antenatal visits may increase premature birth, some of the participants in our study indicated that they noted

a decrease in the number of premature births [26]. This observation is supported by a recent Dutch study [29]. A meta-analysis showed no reductions in preterm births globally, but a lower rate in high income countries [30].

Participants in our study mentioned that they observed an increase in maternal concerns and stress. Maternal stress may be implicated in premature birth in addition to other negative perinatal outcomes. Rondó et al. an association between maternal distress, low birth weight (P = 0.019) and prematurity (p = 0.015) [31]. More work is needed to explore women's lived experiences of pregnancy during the COVID-19 pandemic, the associated reduction in access to antenatal care and the influence that this may have had on maternal perceptions of stress. Moreover, it is important to quantify the numbers of premature births during the various waves of the COVID-19 pandemic and draw comparisons with a similar time frames in other years.

### A perceived increased number of home births

Coxon et al. illustrates that, during the first wave of the COVID-19 pandemic, women were more likely to choose their home, rather than a hospital for childbirth and in the United Kingdom, the NHS released guidelines, stating that home births and midwife-led settings were safest for low-risk women [4]. In our study, almost all of the participants indicated that, in their perception, the number of home births increased during the first wave of the pandemic. Some suggestions were offered for why this might be; women experienced a higher threshold to go a hospital. It was noted by participants that women avoided health care institutions. Another reason mentioned by participants was that during childbirth in hospital, women could only have one companion. Often this was their partner, excluding other support such as family members or doulas. This may have contributed to women making other choices for their childbirth experience. The factors above resulted in a perceived increase in the number of home births. Data for 2020 is not yet available to quantify if there was an actual increase, although it seems probable.

In 2018, 27.9% of all births in the Netherlands were facilitated by community-based midwives, either in the woman's home (12,9%) or as an out-patient (15%). The remaining births (71.1%) were facilitated in hospital, under the supervision of an obstetrician, although the vast majority of these were also attended by hospital-based midwives [32]. Over the last few years, the home birth rate in the Netherlands declined steadily [33]. However, there is increasing evidence that, for low risk women, outcomes for them and their babies are comparable, with lower Apgar scores and fewer unnecessary medical interventions [34].

Participants in this study did not mention that high risk women avoided maternity care during the pandemic. It is possible that they did but that the professionals in this study were not aware of this.

### Improving sense of security and cooperation between care providers

Most participants stated that they were not overly concerned about becoming infected by COVID-19 themselves, although some participants expressed some concerns. For the future, it will be important to ensure that care providers feel safe enough to perform their duties, for example by providing enough PPE for all care providers. Moreover a focus on the emotional safety of maternity care providers is critical, particularly since in our study community-based midwives expressed concerns about a decrease in job satisfaction during COVID-19 [13].

Overall, the obstetricians and hospital based midwives in this study were more positive about the cooperation within their own echelon and across echelons than community-based midwives, although the COVID-19 measures led to some frictions.

### Strengths and limitations

There are several strengths to this study. The diverse nature of the study group and research team and recruitment from all parts of the country enabled the capture of a wide variety of opinions in this study. Another strength is the diversity of the participant group, which created a sample of all relevant categories of maternity care providers in the Netherlands. In addition, this study was conceived and initiated very early on in the pandemic, with interviews taking place at the end of the first wave. This provides a unique insight into the opinions of Dutch maternity care providers in the midst of restriction measures. Finally, we used a rigorous methodology, underpinning the validity of our findings.

There are, however, some limitations as well. It may be that these findings are not applicable to other countries and healthcare systems, because of the uniqueness of Dutch maternity care. However, the measures that the participants outlined in this study, have also been described in other countries [19]. Another possible limitation is that some of the participants were known to the interviewers through personal networks. This may have coloured their comments to a certain extent. In addition, participants might have given socially desirable answers because the interviewers were a midwife and gynaecologist themselves. Considering the variety of opinions that were expressed, this does not appear to have been a major problem.

## Conclusion

This qualitative study explored maternity care providers' experiences with the changes in the organization of maternity care as a result of the COVID-19 pandemic. The pandemic had a major impact on the provision of maternity care. Overall, the participants were positive about many of the measures that were taken and about their proportionality. The participants experienced a strong feeling of unity, an enhanced team spirit and good cooperation during the first wave of the COVID-19 pandemic, although some community-based midwives indicated that the cooperation and resulting friction between different midwifery care practices was suboptimal. This research shows the resilience of the Dutch maternity care system during a public health crisis, but also indicates there are opportunities for improvement. Special attention is needed for the importance of partner involvement, continuity of care, reducing the threshold to seek care and limiting care provider competition. Finally, the COVID-19 crisis has highlighted opportunities for the use of e-Health, reconsideration of medically necessary care and more individualised care. Further research to explore pregnant women's views on the organisation of care during the pandemic is in progress.

## Supporting information

**S1 Appendix. Topic list.**
(DOCX)

**S2 Appendix. Themes and subthemes.**
(DOCX)

## Author Contributions

**Conceptualization:** Suzanne M. Thompson, Janneke T. Gitsels van der Wal, Ank de Jonge, Martine H. Hollander.

**Data curation:** Iris F. Appelman, Suzanne M. Thompson, Lauri M. M. van den Berg, Martine H. Hollander.

**Formal analysis:** Iris F. Appelman, Suzanne M. Thompson, Martine H. Hollander.

**Funding acquisition:** Ank de Jonge.

**Investigation:** Iris F. Appelman, Suzanne M. Thompson, Lauri M. M. van den Berg, Janneke T. Gitsels van der Wal, Ank de Jonge, Martine H. Hollander.

**Methodology:** Iris F. Appelman, Suzanne M. Thompson, Lauri M. M. van den Berg, Janneke T. Gitsels van der Wal, Ank de Jonge, Martine H. Hollander.

**Project administration:** Iris F. Appelman, Suzanne M. Thompson, Lauri M. M. van den Berg, Janneke T. Gitsels van der Wal, Ank de Jonge, Martine H. Hollander.

**Resources:** Ank de Jonge, Martine H. Hollander.

**Supervision:** Suzanne M. Thompson, Janneke T. Gitsels van der Wal, Ank de Jonge, Martine H. Hollander.

**Validation:** Ank de Jonge, Martine H. Hollander.

**Visualization:** Lauri M. M. van den Berg, Janneke T. Gitsels van der Wal, Ank de Jonge, Martine H. Hollander.

**Writing – original draft:** Iris F. Appelman.

**Writing – review & editing:** Iris F. Appelman, Suzanne M. Thompson, Lauri M. M. van den Berg, Janneke T. Gitsels van der Wal, Ank de Jonge, Martine H. Hollander.

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
