## [Decision Letter · Decision Letter 0]

17 Jun 2021

PONE-D-21-05312

It was tough, but necessary. Organizational changes in a community based maternity care system during the first wave of the COVID-19 pandemic: a qualitative analysis in the Netherlands.

PLOS ONE

Dear Dr. Appelman,

Thank you for submitting your manuscript to PLOS ONE. After careful consideration, we feel that it has merit but does not fully meet PLOS ONE’s publication criteria as it currently stands. Therefore, we invite you to submit a revised version of the manuscript that addresses the points raised during the review process.

We look forward to receiving your revised manuscript.

Kind regards,

Rakhi Dandona

Academic Editor

PLOS ONE

Journal Requirements:

Additional Editor Comments (if provided):

Reviewers' comments:

Reviewer's Responses to Questions

**Comments to the Author**

1. Is the manuscript technically sound, and do the data support the conclusions?

Reviewer #1: Yes

Reviewer #2: Yes

2. Has the statistical analysis been performed appropriately and rigorously? 

Reviewer #1: N/A

Reviewer #2: N/A

3. Have the authors made all data underlying the findings in their manuscript fully available?

Reviewer #1: Yes

Reviewer #2: Yes

4. Is the manuscript presented in an intelligible fashion and written in standard English?

Reviewer #1: Yes

Reviewer #2: Yes

5. Review Comments to the Author

Reviewer #1: I have had the opportunity to review your manuscript and have the following commentary:

GENERAL:

ABSTRACT:

- I think you are suffering from methodological pluralism here - see comments on methods.

- If you are using themtic analysis by Braun & Clarke, I would urge you not to use words such as emerge/emergence, but rather generated or identified when discussing themes.

INTRODUCTION:

- I would first stylise COVID-19 as: 'The Coronavirus 'SARS-CoV-2' or COVID-19 pandemic...' in the first line.

- Page 3, line 56, 'on it's arrival' sounds as if COVID came for a holiday, and detracts from the gravity of the virus - perhaps watch your language here to ensure academic style is maintained throughout.

- Page 3, lines 70-72 - it would be helpful for the international readership of PLoS One, for you to explain what the number of prenatal visits was reduced from, to make only seven left over.

- Otherwise very well written and informative.

METHODS:

- Not sure what you mean by 'affinity with maternity care'.

- Your description of the waag communal weighing is delightful and really helps contextualise the Dutch setting.

- You may have to clarify what you mean by 'residents' - as this will have different meanings depending on the country. For example, whilst it is occasionally use in the UK for junior medical staff, the more common use of residents is as a synonym for inhabitants or occupants.

- Is you sampling plan table representative of the actual participants or who you hoped to interview. This needs to be clarified.

- Your data analysis section is where I have an issue. You appear to have what I would deem, 'methodological pluralism' whereby you are drawing upon a mixture of analytical methodologies to explain what you may have actually done. You suggest you are using a constructivist approach to thematic analysis, but you cite methodology from Charmazian Grounded Theory and discuss open, selective, and axial coding which are all reminiscent of Grounded Theory. I would suggest you either align your methodology to Grounded Theory (if you are wanting to develop a theory from your data) OR align your methodology to Thematic Analysis (if you are wishing to identify the broad base themes present in your data). I would also avoid use of terminology such as code-tree which makes it sound as if you have used a codebook, which again would not be appropriate or relevant for Grounded Theory or Thematic Analysis.

- Please could you ecxplain the reasoning as to why ethical approval was deemed not necessary.

RESULTS:

- Page 8, Line 171 - remove comma after word interviews.

- What type of data saturation did you use? On what principles did you judge data saturation? Francis et al., 2010 - what is an adequate sample size OR Guest et al., 2006 - How many interviews are eniygh OR Malterud et al., 2015 - Sample size for interviews OR Saunders et al., 2018 - Saturation in qualitative research OR Vasileiou et al., 2018 - Interviews study sample size - would all be good, but different ways of explaining data saturation - so it would be good to know where you positioned yourself.

- If you are using Thematic Analysis, refrain from using 'emerge' when discussing themes. Rather, use: identified/generated.

- Your theme names require a bit of interpretive refinement. At the moment they are very descriptive and they need to be worked up a little more to be a bit more refined and interpretive. They should also be memorable rather than basic descriptions. They need a bit more analytic work.

- When you state 'over-arching theme' do you mean this as a Central Organising Concept (in the case of Thematic Analysis) or do you mean this as a Theory (in the case of Grounded Theory). Here it is very unclear as to what analytic methodology you have used and unfortunately, this severely weakens your paper.

- Page 9, lines 179-181 - it is entirely unclear what you mean by 'divided into two themes'. Which themes are you now talking about?

- From reading your results, it appears you could equally write this up as a Thematic Analysis OR a Grounded Theory. Given the steps of coding you have done, and the fact that you appear to have an over-arching theory at the end of your results section (rather than an overarching theme as you have called it, or Central Organising Concept as it would be called in Thematic Analysis), I would recommend you amend your manuscript to describe a Grounded Theory analysis rather than a Thematic Analysis. I would then spend some more time describing what you have called an 'overarching theme', but describe it as an Overarching Theory instead (but you would not present data or quotations under the Overarching Theory, so please remove this quotation). This would then echo your title, which makes it instantly look like a Grounded Theory, as you have the Theory in the title already (which made me first think this was a Grounded Theory paper). Your work would be much better suited to a Grounded Theory and I think you would have a stronger manuscript for it if you made these changes. Seek further advice from Silverio et al., 2019 - (Re)discovering grounded theory for cross-disciplinary

qualitative health research; for the steps required and terminology used in Grounded Theory that you will need to imbed into your paper, and systematically remove all reference to thematic analysis throughout the paper.

- Results are otherwise well written with approrpiate amount of quotations (but it reads much more like a Grounded Theory than a Thematic Analysis, so you would be wise to describe it as such).

DISCUSSION:

- Very well written discussion and I like the comparisons between countries as well - this gives you paper context and relevance outside of The Netherlands.

- It is a bit long and could be tightened/sharpened to ensure you keep hold of the reader's interest.

- I am unsure as to why you have so strongly focused on preterm birth in your discussion, which was not overly evident throughout the rest of your paper - possibly consider toning down?

- Page 21, line 484, it appears to be the only time you cite a reference using the authors name, which makes it slightly disjointed and not uniform. Consider revising sentence to omit the name, but retain the citation as a number. You do this again on lines 501, 506, 509, 515, 524, 542, 543 - do consider revising.

- Page23, line 525 - I am assuming you mean COVID-19 not COVID-29!?

- I think you have underplayed your strengths. The fact you did this in the early phases of the pandemic to give us a snapshot of the Dutch context; the fact you have used a rigorous methodology (although you have not written it up exact enough as yet); the diversity of your participants.

- I would disagree with your point about a pre-determined topic list not being a major issue, especially as you detail above that other questions were introduced into interviews if they arose, thus meaning your topic guide was not suitable enough. It's okay to have limitations - we all do - just be a little more honest about them.

CONCLUSION:

- Great - nice and concise!

REFERENCES:

- Ref 2, you have spelt Fernandez Turienzo incorrectly.

- The following references may be of help/interest to you:

Fryer K, Delgado A, Foti T, Reid CN, Marshall J. Implementation of obstetric telehealth during COVID-19 and beyond. Maternal and child health journal. 2020 Sep;24(9):1104-10. DOI: 10.1007/s10995-020-02967-7

Aziz A, Zork N, Aubey JJ, Baptiste CD, D'alton ME, Emeruwa UN, Fuchs KM, Goffman D, Gyamfi-Bannerman C, Haythe JH, LaSala AP. Telehealth for high-risk pregnancies in the setting of the COVID-19 pandemic. American journal of perinatology. 2020 Jun;37(8):800-8. DOI: 10.1055/s-0040-1712121

Madden N, Emeruwa UN, Friedman AM, Aubey JJ, Aziz A, Baptiste CD, Coletta JM, D'Alton ME, Fuchs KM, Goffman D, Gyamfi-Bannerman C. Telehealth uptake into prenatal care and provider attitudes during the COVID-19 pandemic in New York City: a quantitative and qualitative analysis. American journal of perinatology. 2020 Aug;37(10):1005-14. DOI: 10.1055/s-0040-1712939

Reviewer #2: • Overall: This is an interesting piece of research and very relevant in terms of how maternity care can better respond to unprecedented health crisis as well for what can be learned from the COVID-19 response. There is a need to reform practices and build preparedness for future possibilities of crisis like the current pandemic. It is also relevant in presenting the perspectives and voices of frontline healthcare providers, often missed in policy formulations and recommendations.

• Title: The title reads ‘It was tough but necessary’ , obviously taking from what emerged as their overarching theme. However, in reading the findings, it seems there are varied opinions that suggest that not everybody felt the measures were necessary. For example, the community-based midwives tend to have a different opinion on not being allowed to visit even when the clients did not have symptoms or the fact that in the north, the respondents did not agree with the timing or felt the measures were too drastic. You may revisit your title

• Methods:

o The authors mention the word ‘reflexivity’ but do not demonstrate how this was applied and how it may be reflected in study findings, except to say who the researchers were.

• Sampling

o They do not mention what sampling strategy was used and how did they arrive at their sample size

o Although the authors provide a link - it would be useful if they provide a sense of the sample of the mix-methods study - so that one could locate this study in the larger context and get a sense of how the qualitative sample matches up to the overall sample.

o They state that the sample was representative. What is not clear is what was the overall universe for this to be representative of The paper suggests that all hospitals and community-based midwifery practices are uniform across the country. It may be so. However, if it is not, then the authors may want to provide some sense of where the providers are located, as clearly positive and negative perceptions on the measures would vary upon the type of facility and the context.

o In the study design the authors mention – “that they wanted to ‘weigh’ the measures taken against the COVID-19 pandemic, to establish if there was balance between the measures taken and the impact that they had”. This is a qualitative study - how do they weigh the measure and that too without having a real sense of the impact – perhaps they could consider rephrasing.

• Need for a context: The paper will gain immensely if the authors could provide a brief context section that gives the broad outline of the Dutch health system (that they mention as ‘unique’ towards the end but don’t say how or why); what was the exposure to the pandemic in Netherlands and what was the broad response strategy of the health system; and how it affected normal functioning. This will help the findings in the context – for example, why were the changes in the maternity care protocol made? It could also help explain some of the quotes where the hospital midwifes state that they had very high burden (line 238). Also, in a quote it states that it escalated fast, the number of cases kept increasing etc. While this is in a quote, but we have no background to interpret the quote against.

• Findings and Analysis

o Given that the sample comprises 3 types of respondents, all of whom vary either in terms of their specialisation, position and/or location, it would have been useful to locate their responses in specific contexts. The authors mention several times that community-based midwifes had a different experience in terms of the restrictions on meeting in-person, or the non-uniform implementation of the national guidelines etc. However, they don’t state why this was the case - how did their specific practice of being community-based midwifes vary from that of those who were hospital-based and how much of their perception is related to the specificities of their location and practice.

o There are some unsubstantiated statements made in specific sections. For instance, in line 183 they mention the word ‘fear’ and concern and the quote talks about ‘pity’ and concern – which is not the same as fear. Similarly, they mention that this ‘negatively impacted their wellbeing’ (line 183-184), but they don’t say how? In the absence any other data, how is this to be established?

o The authors may want to rephrase what appear to be inconsistencies. In line 189-190 they say impact of COVID-19 was not noticeable in the maternity wards but later they mention that respondents talked of the escalating numbers and work burden

o In line 194-195, the authors talk of the impact on outpatient delivery and intrapartum care, but in the absence of an overall context it is not clear why it was done, what the scale of this was and thus to know what the potential impact of this might be. If the responses could be reflected against what the national scenario was on these issues, the significance of this may be clearer.

o Similar inconsistency is seen in line 204-206. “There was a difference of opinion between maternity care providers about the proportionality and timeliness of the measures that were implemented. Overall, most participants were content with the proportionality and timeliness of the measures”. The two sentences seem do not reconcile. There are different categories of respondents and different locations – could these differences arise because the category they belong to? If so, can we club them together when we say, ‘overall most participants were content’? also the text that follows does not seem to be consistent with this interpretation.

o In line 217-219 quote, not much is said about the awareness levels and backgrounds of the midwives that may have informed their opinions. For instance, here the statement indicates that she may not be aware that people can be infected and yet be asymptomatic – if aware, it may have lead the respondent to assume that the restriction was justified?

o In line 224, without adequate background and context it is difficult to understand terms like “capacity problem”.

o The authors talk about cooperation and communication as a separate section (291) but they also provide details about cooperation in the hospitals in the previous section (258-organisational effect). They may want to club them together

o Overall, I feel that the findings under each section needs to be structured for the reader to be able to better differentiate the varying perceptions among the different categories /regions and why? In the way it is currently organised, these differences seem embedded in the flow of the narrative and do not clearly standout.

o We are dealing with different types of care providers - how did their different professional positions affect the differences of opinions between them? For example, when you say some maternity providers are not on the same page when it comes to certain policies (in the section on coordination and communication)

o Line 299-300 “There are, however, notable differences between the experiences of obstetricians, residents and hospital-based midwives, and the experiences of community-based midwives.” This comes much later but I felt that this frame needs to be set up early in the findings sections and all the findings reflected along these lines

o In line 323-324, if the task force was seen effective by all then why did the community-based midwifes not communicate that there was variation in midwifery practices contravening the national guideline. This the authors mention created competition and rift. There is some inconsistency here that needs to be addressed.

• Discussions

o The discussion section reads well and is nicely done.

o It discusses the pros and cons of the key issues that have emerged but does not really suggest the ‘then what’ or ‘so what?’ For example, one size does not fit all, and gradation would lead women to shop. What are some of the potential ways in which this issue could be addressed or what policy implication might this have?

o In their discussion the authors may want to refer to examples in other countries where COVID hospitals have been set up, especially on their feasibility. Also, their conclusion on inadequate availability of staff seems to suggest that staff from maternity care were also on COVID duty. Some clarity here would be good.

o The findings lend themselves to the medico-techno discussions on pre-mature birth, which also feels incomplete given the number of variables affecting pre-mature birth.

o Given that the authors look at the perception of the providers – the discussions on personal experience and challenges have not been adequately covered – given that so much has been written on health worker challenges in COVID-19, what kinds of stresses did they face outside of just being scared of the infection? Why did they have to do COVID work? They talk of the ‘burden’, but how did that affect them emotionally and physically and how did it in turn impact their work?

o There are some aspects that emerge only in the discussion e.g. PPEs for provider, Dutch health systems being unique etc, without it being reflected in the findings.

o I felt that differences in the different cadre or ‘echelon’ has to be discussed with some context as well as policy implications. The findings suggest that not only did the community-based midwives experience the measure differently from the hospital-based providers, but the difference between them seem to have a direct bearing on care provision (excess referrals that increased workload in the hospital). You mention that the existing differences was aggravated by COVID-19. Some elaboration on this would help.

o The authors may want to highlight more effectively the main contributions that the paper is making

6. PLOS authors have the option to publish the peer review history of their article (what does this mean?). If published, this will include your full peer review and any attached files.

Reviewer #1: No

Reviewer #2: No

---

## [Author Response · Author response to Decision Letter 0]

9 Aug 2021

We thank the reviewers for their time and effort to critically revise our paper and we believe that this has helped to improve our article. We responded to the comments in the document 'Response to reviewers'.

---

## [Decision Letter · Decision Letter 1]

4 Nov 2021

PONE-D-21-05312R1It was tough, but necessary. Organizational changes in a community based maternity care system during the first wave of the COVID-19 pandemic: a qualitative analysis in the Netherlands.PLOS ONE

Dear Dr. Appelman,

Thank you for submitting your manuscript to PLOS ONE. After careful consideration, we feel that it has merit but does not fully meet PLOS ONE’s publication criteria as it currently stands. Therefore, we invite you to submit a revised version of the manuscript that addresses the points raised during the review process.

We look forward to receiving your revised manuscript.

Kind regards,

Rakhi Dandona

Academic Editor

PLOS ONE

Journal Requirements:

Reviewers' comments:

Reviewer's Responses to Questions

**Comments to the Author**

1. If the authors have adequately addressed your comments raised in a previous round of review and you feel that this manuscript is now acceptable for publication, you may indicate that here to bypass the “Comments to the Author” section, enter your conflict of interest statement in the “Confidential to Editor” section, and submit your "Accept" recommendation.

Reviewer #1: All comments have been addressed

Reviewer #2: (No Response)

2. Is the manuscript technically sound, and do the data support the conclusions?

Reviewer #1: Yes

Reviewer #2: Partly

3. Has the statistical analysis been performed appropriately and rigorously? 

Reviewer #1: N/A

Reviewer #2: N/A

4. Have the authors made all data underlying the findings in their manuscript fully available?

Reviewer #1: Yes

Reviewer #2: Yes

5. Is the manuscript presented in an intelligible fashion and written in standard English?

Reviewer #1: Yes

Reviewer #2: Yes

6. Review Comments to the Author

Reviewer #1: I would like to thank the authorship team for working hard to make revisions in-line with my comments and those of the other peer reviewer.

Overall, I am happy with the revised manuscript, but I have two outstanding issues - firstly I would still counsel you against presenting data under the central organising concept, and secondly where you have stated 'obstetrical resident', I think this can be renamed 'obstetric resident'.

Reviewer #2: Thank you for attending to the comments and clarifying some of them

Overall,

- I still feel that the piece could be sharper and more tightened – some data repeats itself (like women were not allowed to be accompanied by their husband or doulas)

- Some of the sub-themes are inconsistent and does not say much like ‘differing view’ while one clearly states the subject of the specific content – reluctance to seek care

- I am still not sure what the unique contribution of the study was – what is that this study helps highlight that is not available from the larger study – the only uniqueness to me is the timing of the study.

- The lessons learnt section overlaps a lot with the discussion section

Minor specific points

- Lines 117-126 – You cite a lot of literature on maternity care provider during COVID-19 but you don’t say how your study is adds to the existing literature – what is different in your study that has not already been looked at

- Technically – qualitative samples cannot be representative. As far as I am concerned representative is more applied to quantitative survey – where the quantum of sample is representative of the larger category of those being sampled. That can never be the case. In qualitative studies its purposive and attempts to include all relevant categories in the sample – like has been done for this study.

- I am not sure if the term ‘over sampled; is being used correctly in line 166-168. What the lines suggest to me that you included a higher sample of midwives and obstetricians because of the importance of their role. Oversampled as far as I know is used when you in your initial sampling you include a higher number than required so that even after non-response you would still get your required sample.

7. PLOS authors have the option to publish the peer review history of their article (what does this mean?). If published, this will include your full peer review and any attached files.

Reviewer #1: No

Reviewer #2: No

---

## [Author Response · Author response to Decision Letter 1]

7 Dec 2021

Dear Editor, 

We want to thank you again for reviewing our manuscipt.

---

## [Editor Report · Decision Letter 2]

9 Feb 2022

It was tough, but necessary. Organizational changes in a community based maternity care system during the first wave of the COVID-19 pandemic: a qualitative analysis in the Netherlands.

PONE-D-21-05312R2

Dear Dr. Appelman,

We’re pleased to inform you that your manuscript has been judged scientifically suitable for publication and will be formally accepted for publication once it meets all outstanding technical requirements.

Kind regards,

Rakhi Dandona

Academic Editor

PLOS ONE
---

## [Editor Report · Acceptance letter]

1 Mar 2022

PONE-D-21-05312R2 

It was tough, but necessary. Organizational changes in a community based maternity care system during the first wave of the COVID-19 pandemic: a qualitative analysis in the Netherlands. 

Dear Dr. Appelman:

I'm pleased to inform you that your manuscript has been deemed suitable for publication in PLOS ONE. Congratulations! Your manuscript is now with our production department. 

Kind regards, 

on behalf of

Dr. Rakhi Dandona 

Academic Editor

PLOS ONE